# Intra-model Ensemble Learning For Single Sample Test-Time Adaptation

## Abstract

Test-Time Adaptation (TTA) problems involve adapting pre-trained models to new data distributions in testing time, with access to only model weights and a stream of unlabeled data. In this work, we present Intra-model Ensemble Learning (IEL), a method of actively adapting a set of pre-trained classifiers to process distribution-shifted data one sample at a time without access to labels. IEL produces higher Ensemble classification accuracy and better individual classifiers that continue to improve and learn from each other during inference, which is why we call our method Intra-model Ensemble Learning. Specifically we minimize the cross-entropy between the classifier output that has the highest predicted probability for the majority-voted class (a high confidence softmax) and all other classifiers. The majority-voted model that all others learn from may change from sample to sample, allowing the group to collectively and continuously learn from each other during testing time. Our method uniquely optimizes all trainable parameters in each model and needs only a single sample (instead of a batch) for adaptation, which is a more practical setting of inference for real-world applications. In our extensive experiments, using sets of independently pre-trained base classifiers with distinct architectures, we show that IEL can reduce generalization error significantly better than base Ensemble model for classification tasks on corrupted CIFAR-10, CIFAR-100, and ImageNet datasets while minimizing the entropy of model outputs.

## 1 Introduction

Modern classifiers can reliably produce accurate predictions when training and testing data are drawn from identical distributions. When there is a *distribution shift* between the training (source) and testing (target) data, i.e. the sets of source and target samples are drawn from distinct distributions, then the recognition performance of many high variance classifiers significantly diminishes (Quionero-Candela et al., 2009; Liang et al., 2023). Distribution shifts are common in many real-world applications such as when sensors in medical imaging equipment degrade enough to distort readings (Karani et al., 2021; Hu et al., 2024). In such cases model deployment may still be necessary, even though available pre-trained models cannot achieve optimal accuracy on testing data due to distribution shifts. In this work, we present a method for adapting a set of independently pre-trained classifiers as an ensemble to distribution-shifted data one sample at a time during inference, assuming that the initial training data for all classifiers was similarly distributed, e.g., identical training sets.

Given a set of pre-trained models, we propose to use the softmax of the member model which has the highest predicted probability for the majority voted class as a soft target (e.g., pseudo-label) to apply backpropagation to all other member models. Since the model selected by our dynamic majority vote scheme may change from sample to sample, member classifiers in the ensemble will *dynamically* learn from each other and continue to adapt during inference until they agree with the predictions of the other models in the set. This marks a significant difference from most methods in Ensemble Learning, such as bagging and stacking, which produce a single strong ensemble classifier from base classifiers in a *static* way, without changing any of the individual base classifiers. Hence, we title our approach as *Intra-model Ensemble Learning* (IEL), inspired by mutually beneficial human collaboration that not only yields superior overall outcomes but also improves the ability of the participants involved.

This paper focuses on Intra-model Ensemble Learning in the context of Test-Time Adaptation (TTA) problems, where we seek to adapt pre-trained models to unlabeled and distribution shifted data. Access to training data may be infeasible during the testing stage, such as in confidential medical or financial domains, and adapted predictions may be required on only a single sample. To address these restrictive cases, for IEL we assume that unlabeled inference samples comes in only one at a time (Khurana et al., 2022; Valanarasu et al., 2024), unlike some TTA settings that require batches of multiple samples for adaptation. When only one sample is available for adaptation, estimation of testing data distributions can pose difficulties for many existing Test-Time Adaptation methods. For example, methods like TENT (Wang et al., 2021), whose performance in part relies on accurate estimations of batch statistics on incoming unlabeled data, become ineffective with access to only a single sample per batch. Besides filling this gap in Ensemble learning research, our IEL approach is more versatile and flexible in many critical domains such as medicine, finance, safety, and security where timely processing is required and reliable batch statistics are hard to acquire.

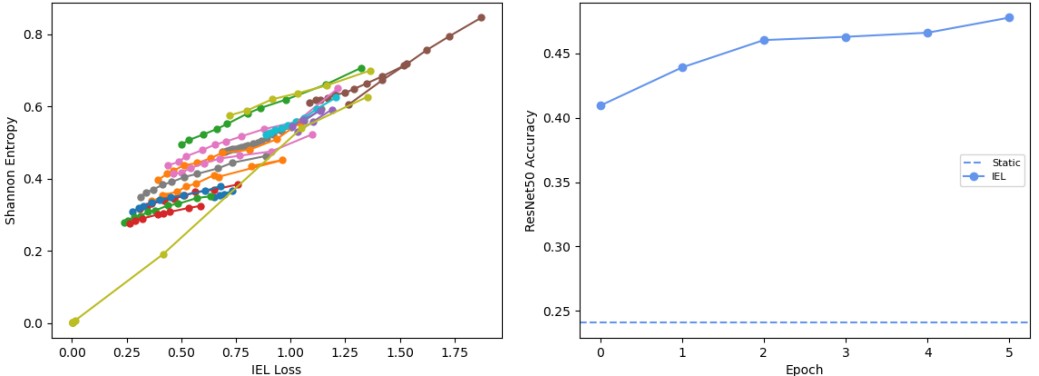

Figure 1: Predictions with lower IEL loss also have lower entropy on ImageNet-C. The coloring of points and lines corresponds to 15 different corruption types.

Figure 2: Accuracy of ResNet50 classifier during IEL on the Glass Blur corruption type from ImageNetC.

In the general Test-Time Adaptation setting where access to multiple samples per batch is allowed, many lines of work have found success with minimizing the Shannon Entropy of classifier outputs (Niu et al., 2023; Jing et al., 2022). This approach optimizes model outputs to achieve high confidence as measured by the Shannon Entropy. Since our choice of soft target has maximal probability for some class it is likely to also have a low Shannon Entropy (high confidence). Compared to using one-hot "hard" targets which correspond to the majority voted class, a soft target also provides more information about relative similarities between classes as shown in the field of Knowledge Distilling (Hinton et al., 2015; Arazo et al., 2020). Still, models are known to be confidently incorrect at times, and so choosing a maximum probability soft target does not always correspond to the correct prediction. By defining our chosen soft target to agree with the majority voted prediction we attempt to leverage the robustness of ensembles to ensure that the target is likely correct.

If the ensemble prediction is incorrect, then allowing it to supervise member models could lead to erroneously optimizing strong models on incorrect predictions. This phenomena results in cascading error accumulations known as *catastrophic forgetting* in the literature on Test Time Adaptation (Wang et al., 2022; Liang et al., 2023). Although the Shannon Entropy has been shown to pose issues as an optimization signal when non-stationary distributions are encountered (Lee et al., 2024), we find that IEL can effectively minimize the Shannon Entropy of model softmax outputs (Figure 1) as a side effect while improving the classification accuracies of member models in stationary TTA settings (Figure 2).

Collaboration between skilled human learners with diverse backgrounds is an effective way that new knowledge is generated when ground truth information is unavailable. Collaborating learners should be skilled so that their contributions to the group are sound, and the requirement that the set

of learners be diverse is premised on the idea that the individual contributions of a single learner may shine light on things not known by another. For example, cutting-edge scientific research on a challenging problem with no known solution is similar to the testing phase of classification in machine learning where ground-truth labels are not available. If individual researchers can not make progress, naturally they will talk, exchange ideas, and collaborate with other researchers(sometimes from different disciplines).

In this collaborative ensemble learning setting, individual learners inspire and gain knowledge from each other in a way that is similar to human collaboration. If successful, the outcome is not only a scientific breakthrough, but also better researchers/learners, which can be more critical for the progression of a scientific field than a single solution in the long run. We aim to emulate the benefits that a group of learners may get from collaboratively solving a problem together by learning from their combined output. By minimizing the cross-entropy distances between a majority voted model output and all others on new samples, we minimize the diversity of the ensemble (we force models to agree with each other) in a way that facilitates generalization of the source training domain to the new testing domain.

**Our Contributions**

- This work proposes diversity as a new optimization signal for classification in the ensemble learning setting, which complements the primary label and loss signal. Since the diversity signal does not require labels, it can be applied during both the training and testing phases. In the training phase, it can supplement label information to improve training performance and efficiency. In testing, it can guide a model to adapt itself to the distribution of test data, which makes our approach a natural choice for Test Time Adaptation and continual learning.

- A standard ensemble model generates predictions through simple mechanisms such as voting or averaging, while all individual models in the ensemble remain fixed. Our approach performs continuous optimization on all member models and creates an upward spiral effect for continual learning, which is why we call our approach Intra-model Ensemble Learning.

- This work considers single sample setting for TTA, which is more restrictive than TTA settings that require a batch of multiple samples for adaptation.

- With precise details to explain how two heads are better than one, this work represents a solid step forward in understanding human collaboration and ensemble methods, and their interrelated connections.

## 2 RELATED WORK

### 2.1 ENSEMBLE LEARNING

Ensemble learning combines the outputs of multiple member models to generate more accurate predictions than any of the constituent models alone (Dietterich, 2000). Popular methods include bagging, boosting, stacking, voting, and Bayesian model averaging (Breiman, 1996; Freund & Schapire, 1997; Schapire & Singer, 1999). Since the objective in the standard Ensemble Learning setting is to only achieve better predictions, constituent models remain unchanged.

A simple way to combine the predictions of models on an arbitrary input sample is to choose the class that was most frequently predicted by the member models, a process referred to as Majority Voting. It is well known that ensembles with high diversity (made from models which make independent errors) have improved prediction robustness, where the individual model errors cancel each other out during the model combination process (Wang & Ji, 2023). Despite the common use of ensemble learning in modern machine learning methods, the presence of diversity among ensemble members is often thought of as a "black box" for improving classification accuracy. Still, many works on ensembles indicate that to get a strong ensemble, diversity should be present among members.

We can view the cross-entropy between member models and the majority voted model as a rudimentary estimate of the diversity in an ensemble, and by minimizing it we force members to agree with the majority voted model. Recent work has shown that the diversity of an Ensemble directly relates to the degree of error reduction that the ensemble provides over its member models (Wood et al., 2023). A natural counter-example to this claim is to form an ensemble from member models that are each an

identical copy of one perfect model and its pre-trained weights such that the copied model achieves 100% classification accuracy on validation data. Such an ensemble would have no diversity among its identical members, but the overall ensemble would retain perfect accuracy, perhaps contingent upon the choice of model combination method used. As such, we expect that is possible to reduce diversity while also reducing generalization error in an ensemble.

## 2.2 TEST-TIME ADAPTATION (TTA)

The foremost aim in TTA is to improve the performance of a source pre-trained model on a target domain when the source and target are not identically distributed (Zhao et al., 2023). Many lines of work in TTA have found success with minimizing the Shannon Entropy of a model's softmax output as a signal in the absence of labels (Gui et al., 2024; Zhang et al., 2022). Recently several new studies have shown that reducing prediction entropy can result in problematic model behaviors, such as leading models to produce over confident predictions (poor calibration) and inducing catastrophic forgetting when spurious correlation shifts are encountered. (Lee et al., 2024; Yoon et al., 2024).

Methods like TENT are able to mitigate these difficulties by re-estimating batch statistics on new data, but not all TTA use cases allow access to more than a single sample at a time. In the setting where only a single sample is available at a time, EATA proposed to use weight-averaged and augmentation-averaged predictions for entropy optimization, while ROID explicitly depends on forming an ensemble from prior versions of an adapted model to facilitate generalization to new domains (Marsden et al., 2023; Niu et al., 2022). All of these methods are able to adapt a single model at a time to new domains, while we choose to focus on the simultaneous adaptation of multiple models to new data through the ensemble combined soft target. Although our method minimizes the Shannon Entropy of model predictions as a side effect (Figure 1), we rely on the inherent diversity in a set of independently pre-trained models to make our models more accurate (Figure 2).

COTTA avoids catastrophic by stochastically restoring small subsets of the model weights to their original values during each iteration, whereas we make no attempts to avoid catastrophic forgetting (Wang et al., 2022). We aim to show the viability of true ensemble based approaches in TTA settings using member models that have been independently pre-trained. The introduction of the Shannon Entropy as an optimization signal in TTA problems was motivated by the general idea of confidence optimization. The Shannon Entropy can be interpreted as measuring the uncertainty present in a softmax layer, so entropy minimization approaches effectively minimize the uncertainty in model predictions. Since we minimize a statistical distance between member models, we expect that IEL also minimizes prediction uncertainty.

This problem can be addressed within the context of Knowledge Distillation if the only available representation of the target distribution comes from the outputs of a pre-trained teacher model, and the student model has also been pre-trained on a source distribution shifted from the target. Such distribution shifts pose significant challenges for production machine learning models when source and target data distributions do not exactly align.

## 2.3 KNOWLEDGE DISTILLATION

Knowledge Distillation uses a complex teacher model to supervise a smaller student model. Transferring knowledge to a smaller model reduces computational overheads and allows well-performing models to be deployed to devices with limited resources, like mobile phones and embedded systems. One early line of work found success in training a small student model on test data with hard-targets generated by a pre-trained ensemble (Bucila et al., 2006). However, using one-hot encodings as pseudo-targets effectively forces the student model to never improve beyond the static teacher since it can draw no additional knowledge about the zeroed out classes in the one-hot target.

(Hinton et al., 2015) found that using a teacher model's real-valued softmax output with a temperature parameter $T > 1$ yielded more information rich pseudo-targets which better facilitated generalization in student models as opposed to using generated "hard-targets", that is, one-hot pseudo-targets predicted by the teacher. This follows the idea that given an input sample $x$ that can belong to one of $k$ classes and a teacher model $H : \mathcal{X} \mapsto \mathbb{R}^k$, the probabilities of the incorrect classes relative to each other in the teacher's soft-max contains relevant information about similarities between classes. In our

work we opt to have ensembles produce softmaxes to take advantage of their higher self-information content compared to one-hot pseudo-labels, but do not employ any temperature control.

However, in Knowledge Distillation teacher and student models are usually independent, and learning is one-way and affects only the student model and not the teacher (Gou et al., 2021). Work following this line of thought has found its best results when the teacher outputs used in the distillation process came from the same input data that the teacher was initially trained on. We propose to let the teacher model be an ensemble dependent on the output of the student, so that the distillation process becomes dynamic for both the student and the teacher, and achieves "teaching others teaches yourself" and mechanically shows how "two heads are better than one".

## 3 INTRA-MODEL ENSEMBLE LEARNING

For an input space $\mathcal{X}^{train}$ and $k$-class label space $\mathcal{Y}$, let $h_\theta$ denote a model with weights $\theta$ pre-trained on source training data $\mathcal{D}^{train} = \{(\mathbf{x}_j^{train}, y_j^{train})\}_{j=1}^N \subset \mathcal{X}^{train} \times \mathcal{Y}$. Assume we have a set of $M$ models each pre-trained on $\mathcal{D}^{train}$ and parameterized by weights $\theta_i$ for $i = 1, ..., M$. We do not require that the models have identical architectures, only that that they have softmax outputs, so unless otherwise specified we will assume that any two models with distinct weights also have distinct architectures. Our goal is to adapt the models to test data $\mathcal{D}^{test} = \{(\mathbf{x}_j^{test}, y_j^{test})\} \subset \mathcal{X}^{test} \times \mathcal{Y}$, assuming $\mathcal{X}^{train} \neq \mathcal{X}^{test}$.

Denote the $i$-th classifier's predicted probability of a sample belonging to class $y \in \mathcal{Y}$ as $P(y|\mathbf{x}, \theta_i) = h_{\theta_i}(\mathbf{x})^{(y)}$. On input $\mathbf{x}$ assume that the majority of models predict class $c$, then our soft target is defined as the softmax output of the model which has the highest predicted probability for that class,

$$H(\mathbf{x}) := \underset{\{h_{\theta_i}(\mathbf{x})|i=1,...,M\}}{\arg\max} h_{\theta_i}(\mathbf{x})^{(c)}.$$

Since $H(\mathbf{x})$ uses majority voting to determine the prediction, we expect that it will provide more robust soft targets on the target domain $\mathcal{D}^{test}$ compared to using one fixed model's softmax for adaptation. This would allow weaker models in the ensemble to learn from the group.

For IEL we propose to minimize the summed cross-entropy between all model outputs and the soft target $H(\mathbf{x})$, so our loss is

$$\mathcal{L}(h_{\theta_1}, h_{\theta_2}, ..., h_{\theta_n}, \mathbf{x}) = \frac{1}{M} \sum_{i=1}^M \delta(h_{\theta_i}(\mathbf{x}), H(\mathbf{x})), \tag{1}$$

where $\delta$ is the cross-entropy function. One advantage to using the majority voted softmax for the ensemble output is that the model deemed the strongest by the group should be least affected by backpropagation using the above loss. If the $j$-th model is selected as the output, so $H(\mathbf{x}) = h_{\theta_j}(\mathbf{x})$, then the $j$-th term of the sum in (1) is the cross-entropy of $h_{\theta_j}(\mathbf{x})$ with itself. By the definition of cross-entropy, $\delta(h_{\theta_j}(\mathbf{x}), h_{\theta_j}(\mathbf{x}))$ is simply the entropy of $h_{\theta_j}(\mathbf{x})$, and since $h_{\theta_j}(\mathbf{x})$ was selected to have low-entropy it will make only small contributions to the loss. It is worth noting that by using the cross-entropy in this way we risk overwriting the strong knowledge of the majority voted models. The use of a statistical distance function $\delta$ that is 0 when inputs are identical, like the Kullback–Leibler divergence, will be investigated in future work.

Our loss in (1) is minimized with respect to every model's trainable parameters on each incoming test sample. Each sample is seen only once during an epoch of IEL, so the learning on each sample applies only to future data. Since we work with stationary distribution shifts (when using corrupted data we only stream in samples from one corruption type) this allows us to estimate the target distribution with higher accuracy in later epochs. Some works in ensemble learning have considered the disagreement between member models as a rough measure of prediction uncertainty, so minimizing $\mathcal{L}$ can be thought of as minimizing the uncertainty of the ensemble predictions, perhaps at the cost of model calibration (Sun et al., 2022).

## 3.1 Algorithm

The iterative adaptation procedure for IEL is illustrated in Algorithm 1, and consists of the following 3 phases:

**Initialization** The optimizer collects all trainable parameters from all $M$ models used for IEL. We freeze all batch normalization layer weights so that they can not be updated during testing time.

**Iteration** Each step backpropogates the IEL loss (1) to all models on an incoming sample of data. After backpropogation, a new test sample from the same distirbution is loaded in to repeat the process.

**Termination** No termination is required for IEL, although there are several reasonable conditions for termination that we have not yet unexplored. In future work, we would like to terminate IEL once all model predictions agree with each other.

---

**Algorithm 1** Intra-model Ensemble Learning (IEL)

---

**Require:** Set of $M$ pre-trained source models $S = \{h_{\theta_1}, h_{\theta_2}, ..., h_{\theta_M}\}$, set of unlabeled test-set samples $\mathcal{X}^{test}$ with $k$ classes.
    On input $x \in \mathcal{X}^{test}$, determine majority voted class $c$
    $H \leftarrow \arg\max_{\{h_{\theta_i}(\mathbf{x})|i=1,...,M\}} h_{\theta_i}(\mathbf{x})^{(c)}$          ▷ Set ensemble output
    $\mathcal{L} = \frac{1}{M} \sum_{i=1}^{M} \delta(h_{\theta_i}(\mathbf{x}), H(\mathbf{x}))$          ▷ Compute loss
    Back propagate $\mathcal{L}$ to all models in $S$.

---

We do not include a termination step in our experiments since we want to analyze the long-run behavior of IEL. In Domain Adaptation settings access to the training data and/or the entire set of testing data is allowed, which could also be useful for determining a termination requirement. Once accuracies on the testing set start to diminish, one could terminate IEL to ensure that catastrophic forgetting does not worsen. In Figure 3 we see that the peak average accuracy of an IEL ensemble is reached at 5 training epochs, with future epochs trending towards the static model base line. In some experiments we found that the average model accuracy of the IEL ensemble reduced below the static model accuracy by the final epoch. We will explore techniques for avoiding catastrophic forgetting with IEL in future work.

## 4 Experiments

We evaluate the capability of IEL to adapt an ensemble of independently pre-trained classifiers to test set data. Our code for experiments uses the PyTorch (Paszke et al., 2017) framework for model implementation with training on up to 3 NVIDIA L40S GPUs. We benchmark our performance on three TTA tasks: adapting models pre-trained on CIFAR-10 to CIFAR-10C, on CIFAR-100 to CIFAR-100C, and ImageNet to ImageNet-C (Hendrycks & Dietterich, 2019). Pre-trained weights for CIFAR-10 and CIFAR-100 models were sourced from a public GitHub repository (https://github.com/chenyaofo/pytorch-cifar-models). Pre-Trained weights for ImageNet models were sourced from the PyTorch Vision library. All models used had their batch normalization parameters (if applicable) frozen during the IEL process. This, along with using a batch size of 1, was done to ensure that we are not benefiting from updating batch normalization statistics on new data.

**Datasets** The CIFAR10C, CIFAR100C, and ImageNet-C datasets contain test set images from original CIFAR10, CIFAR100, and ImageNet datasets with 15 common image corruptions applied. Pre-trained models were trained on data from the respective uncorrupted training sets. For CIFAR10C and CIFAR100C we used all 10,000 samples per corruption type for adaptation. Due to computational restraints, for ImageNet we use a class-balanced 7000 sample subset of the data from each corruption type. We separate the data used for each corruption type into two sets, with a 90% split of tuning set samples used for IEL and 10% split of evaluation set samples used strictly for evaluating generalization error on unseen data.

**Optimization** During testing time, we feed a stream of images from a single corruption type to our models and minimize the IEL loss. Each optimization step requires only a single corrupted image for adaptation. After all images from a corruption type have been used, we reset all model weights and start the IEL adaptation process from scratch on the new corruption type. Using only one sample per

batch was empirically found to produce identical performance gains as using multiple samples per batch, so in all experiments we use a batch size of 1.

We apply IEL for several epochs on the corruption types in each dataset. We use a learning rate of 0.001, and a regularization constant $\alpha = 10e^{-11}$ which effectively makes our learning rate even smaller. For each set of models used in our experiments we also track their static accuracies when left unchanged, without applying IEL to them. This serves as our baseline to show much we can improve models over their original capabilities.

Table 1: Highest accuracy improvements (%) over all epochs of an IEL ensemble on CIFAR10-C. For each corruption type we show all model accuracies on the tuning set (left of cell) and the evaluation set (right of cell). Results that show improvement during one of the IEL epochs are bolded.

| Models / Corruptions | Majority Vote accuracy | Average accuracy | resnet20 accuracy | vgg11 bn accuracy | mobilenetv2x05 accuracy | shufflenetv2x05 accuracy | repvgg a0 accuracy |
|---|---|---|---|---|---|---|---|
| Gaussian Noise | -20.56/-17.70 | -16.67/-15.42 | -13.53/-14.40 | -31.01/-27.50 | -7.69/-6.80 | -22.81/-21.60 | -8.30/-6.80 |
| Shot Noise | -4.04/-6.70 | -1.36/-2.28 | **+2.38/+1.20** | -13.64/-13.60 | **+8.06/+6.00** | -7.36/-8.00 | **+3.74/+3.00** |
| Defocus Blur | **+7.67/+9.50** | **+9.45/+10.72** | **+11.26/+12.40** | **+6.84/+8.50** | **+12.38/+13.20** | **+10.68/+12.10** | **+6.49/+7.80** |
| Impulse Noise | -1.74/-1.20 | **+0.50/+0.50** | -0.44/-1.50 | **+0.02/+0.80** | **+2.14/+2.90** | -0.19/-0.90 | **+0.94/+1.20** |
| Glass Blur | **+8.21/+7.00** | **+9.35/+8.88** | **+15.16/+15.60** | **+1.71/+2.00** | **+13.97/+13.20** | **+5.56/+6.50** | **+10.92/+9.00** |
| Frost | **+8.63/+8.80** | **+10.40/+10.42** | **+13.61/+15.80** | **+5.82/+6.30** | **+14.20/+12.20** | **+11.56/+10.60** | **+7.90/+8.60** |
| Zoom Blur | **+14.63/+14.30** | **+16.91/+15.66** | **+20.36/+18.30** | **+10.69/+10.30** | **+21.99/+21.00** | **+18.86/+18.90** | **+12.78/+12.10** |
| Motion Blur | **+10.57/+11.10** | **+12.57/+12.88** | **+15.24/+15.80** | **+6.04/+5.20** | **+19.18/+20.40** | **+13.82/+15.40** | **+9.01/+9.90** |
| Snow | **+2.69/+4.20** | **+3.88/+3.16** | **+5.26/+5.60** | **+2.76/+2.60** | **+3.89/+5.00** | **+5.19/+3.50** | **+2.79/+1.90** |
| Jpeg Compression | **+3.40/+3.90** | **+4.60/+4.60** | **+6.82/+8.00** | **+1.02/+1.10** | **+7.26/+8.40** | **+4.00/+3.10** | **+4.29/+4.60** |
| Elastic Transform | **+7.27/+9.20** | **+8.73/+8.42** | **+10.76/+9.80** | **+5.20/+7.60** | **+10.98/+10.00** | **+10.16/+11.00** | **+6.74/+5.70** |
| Pixelate | **+7.81/+9.30** | **+9.09/+9.84** | **+14.71/+15.40** | **+2.50/+1.90** | **+11.63/+13.00** | **+10.58/+13.90** | **+6.56/+6.60** |
| Fog | **+4.89/+5.80** | **+6.48/+8.20** | **+5.32/+7.70** | **+9.04/+12.20** | **+6.62/+8.50** | **+8.69/+12.10** | **+3.16/+2.70** |
| Contrast | **+1.91/+2.10** | **+5.24/+4.66** | **+2.40/+1.80** | **+11.22/+10.50** | **+5.33/+4.90** | **+9.46/+9.20** | -0.74/-1.00 |
| Brightness | **+0.56/+0.20** | **+1.11/+0.86** | **+1.29/+2.50** | **+1.06/+1.60** | **+1.24/+1.40** | **+2.07/+1.60** | **+0.76/+0.80** |

Table 2: Highest accuracy improvements (%) over all epochs of an IEL ensemble on CIFAR100-C. For each corruption type we show all model accuracies on the tuning set (left of cell) and the evaluation set (right of cell). Results that show improvement during one of the IEL epochs are bolded.

| Models / Corruptions | Majority Vote accuracy | average accuracy | resnet20 accuracy | vgg11 bn accuracy | mobilenetv2x05 accuracy | shufflenetv2x05 accuracy | repvgg a0 accuracy |
|---|---|---|---|---|---|---|---|
| Gaussian Noise | -10.30/-10.00 | -8.36/-8.32 | -6.86/-7.40 | -14.89/-14.80 | -6.09/-6.70 | -7.44/-6.50 | -6.50/-6.20 |
| Jpeg Compression | **+3.93/+4.20** | **+4.36/+4.80** | **+6.26/+6.30** | **+2.58/+1.90** | **+5.87/+7.80** | **+3.54/+5.20** | **+4.68/+5.60** |
| Elastic Transform | **+7.56/+9.00** | **+7.70/+7.94** | **+8.94/+10.60** | **+6.11/+6.90** | **+8.40/+8.50** | **+9.10/+9.60** | **+7.04/+6.90** |
| Motion Blur | **+13.07/+13.60** | **+13.30/+12.40** | **+14.26/+14.70** | **+10.02/+10.30** | **+14.81/+13.60** | **+14.51/+14.30** | **+13.16/+11.90** |
| Brightness | **+1.93/+2.10** | **+2.30/+2.06** | **+1.74/+2.10** | **+2.27/+2.00** | **+2.48/+2.40** | **+2.97/+3.40** | **+2.43/+1.30** |
| Snow | **+3.17/+3.50** | **+4.26/+3.98** | **+4.48/+6.20** | **+3.49/+4.90** | **+5.06/+5.00** | **+5.94/+4.60** | **+3.03/+5.10** |
| Contrast | **+3.19/+4.00** | **+4.27/+5.62** | **+3.56/+6.40** | **+6.79/+6.70** | **+5.43/+5.90** | **+5.99/+7.50** | **+0.70/+1.70** |
| Defocus Blur | **+9.19/+9.80** | **+9.89/+10.08** | **+10.49/+12.80** | **+6.67/+7.40** | **+11.59/+12.30** | **+12.37/+11.40** | **+9.23/+9.70** |
| Shot Noise | -13.74/-11.80 | -10.91/-10.26 | -8.42/-7.40 | -19.24/-18.60 | -9.00/-8.80 | -9.68/-9.70 | -8.21/-6.80 |
| Glass Blur | -6.29/-6.20 | -4.30/-4.44 | -3.14/-4.70 | -6.91/-7.70 | -3.36/-3.30 | -2.80/-2.70 | -5.28/-3.80 |
| Fog | **+4.59/+5.10** | **+5.47/+5.68** | **+4.68/+4.90** | **+6.78/+8.10** | **+5.18/+6.40** | **+7.46/+7.50** | **+3.98/+4.30** |
| Pixelate | **+14.32/+15.80** | **+14.01/+14.66** | **+19.41/+21.10** | **+3.37/+5.60** | **+17.08/+18.50** | **+15.67/+13.60** | **+14.80/+17.10** |
| Frost | **+7.41/+9.70** | **+7.73/+8.46** | **+8.71/+10.50** | **+4.02/+5.70** | **+10.71/+11.40** | **+8.77/+10.70** | **+6.77/+8.10** |
| Zoom Blur | **+12.78/+14.60** | **+13.26/+12.96** | **+14.78/+15.30** | **+9.71/+6.20** | **+14.28/+14.10** | **+15.60/+16.00** | **+11.96/+14.50** |
| Impulse Noise | -15.67/-15.70 | -10.74/-11.16 | -11.46/-12.60 | -4.40/-5.00 | -12.32/-12.00 | -13.22/-13.50 | -12.28/-12.70 |

Table 3: Highest accuracy improvements (%) of an IEL ensemble over a static ensemble on ImageNet-C. For each corruption type we show all model accuracies on the tuning set (left of cell) and the evaluation set (right of cell). Results that show improvement during one of the IEL epochs are bolded.

| Models / Corruptions | majority vote accuracy | average accuracy | resnet101 64x4d accuracy | resnet152 accuracy | resnet101 accuracy | resnet50 accuracy |
|---|---|---|---|---|---|---|
| defocus blur | **+5.97/+7.43** | **+8.58/+7.86** | **+3.59/+4.43** | **+9.08/+8.57** | **+10.05/+10.14** | **+11.62/+9.86** |
| glass blur | **+17.78/+17.71** | **+19.51/+17.64** | **+15.0/+13.86** | **+20.1/+19.0** | **+19.24/+15.0** | **+23.7/+22.71** |
| motion blur | **+13.11/+13.71** | **+16.2/+13.75** | **+11.43/+12.71** | **+15.83/+13.14** | **+17.83/+14.29** | **+19.86/+16.57** |
| zoom blur | **+15.37/+17.86** | **+15.6/+15.89** | **+11.79/+12.86** | **+17.65/+18.14** | **+18.22/+18.0** | **+16.68/+16.0** |
| contrast | **+6.05/+7.14** | **+8.62/+6.89** | **+4.59/+3.86** | **+8.35/+7.14** | **+10.03/+8.14** | **+11.83/+9.71** |
| elastic transform | **+8.48/+7.57** | **+10.85/+8.11** | **+7.13/+7.14** | **+10.92/+5.86** | **+12.08/+10.14** | **+13.29/+10.0** |
| pixelate | **+5.44/+7.0** | **+7.56/+5.93** | **+4.21/+4.71** | **+8.4/+7.71** | **+9.1/+7.43** | **+8.78/+5.14** |
| jpeg compression | **+5.54/+7.14** | **+7.6/+5.71** | **+5.32/+5.57** | **+8.08/+5.57** | **+8.27/+6.0** | **+8.79/+7.14** |
| spatter | **+11.14/+10.86** | **+13.46/+11.43** | **+8.78/+8.86** | **+14.59/+13.86** | **+14.75/+12.14** | **+15.9/+11.86** |
| speckle noise | **+8.83/+9.71** | **+11.93/+9.18** | **+8.16/+9.57** | **+11.37/+9.14** | **+13.79/+11.29** | **+14.46/+7.86** |
| gaussian blur | **+13.14/+12.14** | **+15.42/+11.14** | **+10.41/+9.29** | **+15.81/+10.43** | **+17.63/+12.71** | **+18.11/+13.14** |
| saturate | **+3.38/+4.14** | **+5.35/+2.57** | **+2.02/+1.29** | **+6.14/+5.29** | **+7.06/+3.0** | **+6.17/+2.0** |
| gaussian noise | **+8.19/+8.0** | **+9.91/+9.18** | **+6.6/+6.0** | **+8.65/+9.14** | **+10.73/+8.0** | **+13.71/+14.14** |
| shot noise | **+4.38/+6.0** | **+5.41/+6.11** | **+2.25/+3.0** | **+5.63/+4.86** | **+6.02/+7.0** | **+7.73/+9.57** |
| fog | **+9.14/+9.29** | **+9.38/+10.04** | **+3.7/+4.71** | **+11.37/+14.57** | **+11.67/+10.29** | **+10.78/+10.57** |
| impulse noise | **+7.97/+8.86** | **+9.48/+9.5** | **+6.41/+5.29** | **+8.6/+9.43** | **+10.03/+12.0** | **+12.89/+12.0** |
| frost | **+7.86/+9.14** | **+8.4/+8.89** | **+5.49/+6.57** | **+8.08/+9.86** | **+9.13/+9.29** | **+11.0/+10.0** |
| snow | **+8.21/+6.29** | **+8.7/+8.79** | **+4.24/+5.86** | **+9.56/+10.14** | **+10.11/+10.57** | **+11.17/+10.43** |
| brightness | **+2.29/+3.57** | **+2.4/+2.29** | **+1.02/+0.71** | **+2.89/+2.86** | **+2.87/+4.29** | **+2.84/+1.29** |

## 4.1 EXPERIMENT RESULTS, ANALYSIS AND DISCUSSION

Experiment results on CIFAR10C, CIFAR100C, and ImageNet-C datasets are shown in Table 1, 2, 3 respectively. Here is our analysis and discussion on these results.

**IEL can improve generalization error of pre-trained models.** Table 1 and Table 2 show that IEL successfully increases classification accuracy on both the tuning data used for IEL and the evaluation data used to estimate generalization error. Additionally, for some corruption types we observe that the increase in classification accuracy is higher on the unseen evaluation set data than on the tuning set data, showing that IEL can avoid overfitting in some cases (see resnet152 accuracies on the zoom blur and fog corruption types in Table 3).

**IEL enables models to learn from each other.** Since our optimization objective minimizes the cross-entropy between models in the ensemble, all performance gains observed stem directly from within the ensemble, showing true intra-model ensemble learning. If adaptation is too difficult to show significant performance gains, we find that some models can degrade while others marginally improve (see accuracies on the Impulse Noise corruption type in Table 1). This shows a significant difference from conventional ensemble learning methods that leave individual models unchanged.

**IEL can adapt to a majority of corruption types.** IEL induces catastrophic forgetting on only 3 of the 15 corruption types in CIFAR10C (Gaussian Noise, Shot Noise, and Impulse Blur in Table 1) and 4 of the 15 corruption types in CIFAR100C (Gaussian Noise, Shot Noise, Glass Blur, and Impulse Noise in Table 2). Surprisingly, on ImageNetC no catastrophic forgetting was encountered on any of the corruption types (Table 3).

**IEL reduces entropy and error without dependence on batch normalization layers.** Figure 1 corresponds to the experimental results shown in table 3 for ImageNetC, and shows that the Shannon Entropy is reduced along with our loss in testing time. Figure 2 depicts the accuracy improvements

of a ResNet50 classifier over all training epochs, showing that improvements can still be made if IEL was allowed to run for more epochs.

With these extensive experiments showing IEL's effectiveness on TTA tasks, it is natural to ask if IEL could work in a standard Ensemble Learning setting not involving distribution shifted data. We have performed experiments to evaluate the capabilities of IEL on uncorrupted datasets. We found success with adapting sets of models pre-trained on CIFAR100 and ImageNet to the uncorrupted test sets of both datasets. The average model accuracies per epoch of IEL on the uncorrupted ImageNet test set is shown in Figure 3. It is worth noting that the performance gains are much smaller in Figure 3, we suspect that is because higher performing models have little left to learn from others. Since this paper focuses on TTA, and it is not feasible or optimal to include all details (different architectural components and analysis of loss and entropy) about IEL's performance on uncorrupted datasets, we will report these findings more comprehensively in another paper.

**Limitations and Future Work** Similar to many standard Ensemble Learning methods, IEL is more computationally heavy than TTA methods that employ only one model and do not use backpropagation. However, we believe that it is a cost worth paying as IEL is able to affect all trainable parameters in models and adapt multiple models simultaneously. It would be interesting to analyze the behavior of IEL as we change the number of models in our set, as many works in Ensemble Learning indicate that having larger ensembles can significantly improve the resulting reduction in generalization error of the ensemble output compared to the outputs of the individual models. Additionally, there are many techniques for avoiding catastrophic forgetting that were not employed in IEL, but could be considered in future work, such as replacing the cross-entropy with a statistical distance function that outputs 0 when both input distributions are identical, or stochastically restoring subsets of model weights after backpropogation.

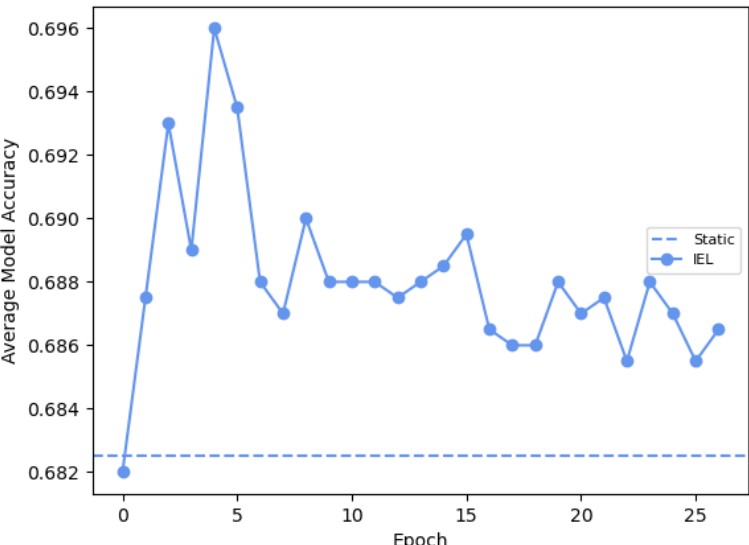

Figure 3: Average model accuracy during IEL training on data from ImageNet test set.

## 5  CONCLUSION

This paper presents our Intra-model Ensemble Learning approach using a new diversity-based optimization signal during inference to improve the classification accuracies of multiple models. This is especially useful for testing phases when the primary optimization signal - ground truth labels - is unavailable. With our IEL approach, member models are optimized and improved like in knowledge distillation and create an upward spiral effect, which not only further improves the overall

prediction performance but also produces better individual models. We validated our method on multiple challenging TTA tasks, which shows significant and consistent improvement over baselines.

**Reproducibility Statement**

To ensure that our findings are reproducible, we will provide a link to our source code after publication. Our code is written to be able to produce results in the form of .csv files for all experimental settings considered in this paper (TTA and non-TTA). We have attempted to convey all necessary steps to recreate our algorithm from scratch in the main content of our paper (sections 3 & 4) should that be necessary. By providing information about the hardware and hyperparameters we used for our experiments, we are sure that anybody attempting to reproduce our results can do so on their machine of choice. The pre-trained weights of source models that were used for IEL will also be provided, and we, the authors, have no affiliation or relation to the owners of those models. Given these resources, anybody can run the experimental results depicted in the main paper, as well as they can make plots from raw .csv files and run our code.

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
