# OpenReview forum: "IEL: Intra-Model Ensemble Learning For Single Sample Test-Time Adaptation"
_ICLR.cc/2025/Conference — Submitted to ICLR 2025_

### Official Review · Reviewer_4Chx · 2024-10-30

**Soundness:** 1
**Presentation:** 2
**Contribution:** 2
**Rating:** 3
**Confidence:** 3

**Summary:**

This work introduces a new method for Test-Time Adaptation (TTA), Intra-model Ensemble Learning (IEL), that optimizes multiple models for ensembled inference. In the IEL framework, the output of the model with the highest probability for the class that received the most votes is set as the pseudo-label, and all other models are optimized to output this pseudo-label via minimizing the cross-entropy. This process minimizes the diversity within the ensemble and aligns the outputs of the models to ensure mutual agreement. Experimental results show the effectiveness of the proposed framework.

**Strengths:**

1. This paper is well written and easy to follow along with their reasoning.
2. The idea of utilizing ensemble methods for TTA is simple, intuitive and easy to adapt.
3. Experiments were conducted on a single-sample setting, which is one of the challenging tasks in TTA.

**Weaknesses:**

1. There is insufficient justification for minimizing the ensemble diversity. Personally, I believe that as diversity increases, performance can be improved in general including TTA since the information is also increases. For example, in [1] which is also referenced in this manuscript, it was shown that increasing diversity can lead to a lower loss. Additionally, the counterexample of ensemble of models with 100% performance (Lines 161–166) is unrealistic and therefore inappropriate for supporting the claim. If there is a large distribution shift and the source-trained models perform poorly on target dataset, reducing diversity may actually have an adverse effect. In conclusion, it remains unclear how reducing ensemble diversity benefits TTA.

2. Due to multiple assumptions, the scope of application of this research is limited. The authors assume there are multiple source-trained models (Lines 230–232), but it is questionable whether this assumption is easily met in practice. Furthermore, the assumption of stationary distribution shifts (Line 265-266) raises concerns about whether IEL would be effective in other TTA scenarios such as online imbalanced label distribution shifts or mixed distribution shifts [2].

3. The experiments conducted do not sufficiently demonstrate the effectiveness of IEL. For example, the authors should include 1) performance comparisons with various previous works on TTA, 2) ablation study on the number of ensemble models, and 3) comparisons of computational costs associated with using multiple models. As mentioned earlier, including experiments across diverse TTA scenarios would also provide a more comprehensive understanding of IEL’s effectiveness.

[1] A unified theory of diversity in ensemble learning
[2] Towards stable test-time adaptation in dynamic wild world

**Questions:**

1. Is there any theoretical or empirical evidence that a minimizing the diversity of the ensemble is beneficial for TTA?

2. In Figure 2, it shows that predictions with lower IEL loss have lower entropy. However, previous study [3], also cited in the paper (Lines 101–105, 173–176), claimed that reducing entropy of output during training can be problematic for TTA. This raises doubts about whether lower IEL loss actually leads to higher TTA performance, and I am curious whether there is any evidence to verify the relationship between IEL and TTA performance.

3. Are there any experimental results that address the weaknesses mentioned (e.g., comparisons with previous studies, ablation study on the number of models, computational cost comparisons, and performance comparisons across various TTA scenarios)?

4. If there are multiple source-trained models, what advantages does IEL offer over an approach that applies conventional TTA methods to each model individually and then performs an ensemble?

[3] Entropy is not enough for test-time adaptation: From the perspective of disentangled factors

---

> ### Author Response · Authors · 2024-11-22
>
> Thank you for taking the time to read our paper and give your review. We are currently working on some of the experiments you've recommended and will update this comment as we get new results.
>
> 1. We are not aware of any evidence to support that minimizing the diversity of the ensemble is beneficial for TTA, but this is also why we found our results interesting. The de facto standard in many ensembling techniques is that diversity is beneficial for the ensemble error as the classification weaknesses of some members can be compensated for by stronger members, however, our work seems to work against that standard since our ensemble can improve while minimizing its diversity.
>
> 2. We are currently studying the general relationship between IEL and TTA performance, particularly in settings where non-stationary distribution shifts are encountered. We will update this comment in the future as we get results.
>
> 3. We do not have the experimental results that you mentioned in your question.
>
> 4. Conventional TTA approaches typically operate on only a single model at a time, and so any adaptations made are inherently dependent on the pre-trained knowledge of a single model. With IEL we are able to depend on the pre-trained knowledge of several models. For example, if the member model in the ensemble with the highest classification accuracy makes an incorrect prediction on a given sample, then other members have the opportunity to correct them, unlike in TTA methods that depend on only a single model.

---

> > ### Comment · Reviewer_4Chx · 2024-11-27
> >
> > I appreciate the response provided by the authors. I have checked their responses and look forward to updates reflecting the new results. Below are my thoughts regarding their responses, which I hope will be considered during the revision of the paper.
> >
> > I agree that reducing the diversity of the ensemble may surprisingly be beneficial in the TTA setting, and this is a noteworthy point. This is particularly intriguing because it challenges the conventional perspective presented in prior studies and offers a new viewpoint. However, to convincingly demonstrate a new perspective that contradicts the established understanding, it is essential to support the argument with more robust theoretical and experimental evidence. Unfortunately, the current manuscript does not provide sufficient evidence for this claim. Additionally, I am curious to know why reducing ensemble diversity is particularly helpful in the TTA setting. Is this claim specific to the TTA context, or could it have more general applicability? The answer to this question would likely influence the direction of future research.

---

### Official Review · Reviewer_SPHA · 2024-10-30

**Soundness:** 2
**Presentation:** 2
**Contribution:** 2
**Rating:** 3
**Confidence:** 4

**Summary:**

The submission describes a method for adapting an ensemble of classifiers to a new data distribution using only unlabeled data. Specifically, the classifiers are trained sequentially soft pseudo-labels, which are the output of the model that has the highest predicted probability for the majority class across classifiers. Experiments are performed on standard datasets (CIFAR10, CIFAR100, ImageNet) with synthetic distribution shifts for an ensemble consisting of 5 pre-trained image classification models.

**Strengths:**

Strengths:
* the studied problem is realistic and relevant
* the propose method makes sense and is simple enough for practical use
* experiments show improved accuracy by the proposed test-time-adaptation

**Weaknesses:**

Unfortunately, the submission has a number of weaknesses.

* incomplete method

The method works iteratively but the submission does not provide a termination condition. This is a not minor issue, as the manuscript makes it sound: without a termination condition, the method cannot be run in practice, and because no labeled data is available at test time, standard techniques, such as checking for a drop of validation accuracy, cannot be applied, either.

* lack of clarity in contribution and scientific comparison to prior work

The submission almost exclusively showcases the proposed method itself, but it does not put the method into context and competition with prior or alternative techniques. This leaves the novelty and relevance of the contribution unclear. Specifically, for a publication at a scientific venue, I expect that the novel aspect of the proposed method is clearly described, contrasting it to ideas and techniques that already existed in previous works. The manuscript does not do a good enough job doing so. As related work it concentrates almost exclusively on recent papers from the application domain of test-time adaptation with deep networks. However, ensemble methods have been studied extensively in machine learning, e.g. [Alaydin, "Introduction to Machine Learning", 2014; Chapter 17], and self-training methods for learning from unlabeled data also have a long history, going back at least to [Fralick, "Learning to Recognize Patterns without a Teacher", 1965], and having emerged many times afters, e.g. in the context of semi-supervised  learning (e.g. Chapelle "Semi-Supervised Learning", 2006) and also test-time adaptation, e.g. [Royer et al, "Classifier adaptation at prediction time", 2015] . Even for adapting ensemble methods self-training was proposed before, e.g. [Ghosh et al, "Self Training with Ensemble of Teacher Models", 2021]. In light of extensive prior work, the manuscript should make clear what the technical novelty of the proposed method is.

* lack of baselines in experiments

The experiments evaluation presents results for the proposed method, but it does not contrast them against prior works. Consequently, the reader only learns that using proposed method often (but not always) works better than not doing test-time adaptation, but if the method is better than already existing methods for test-time-adaptation. It would also be important to see at least a comparison to obvious baselines, such simply self-training each network individually. For the specific choices made, e.g. using the majority vote prediction as target label but the softmax scores of the most confidence classifier, an ablation study would be needed if this is actually useful, or if maybe using hard labels or the ensemble average confidence would work equally well (or better).

* unsubstantiated claims

The manuscript contains factual claims, e.g. about design choices that are not self-evident but also not  substantiated with evidence. Some of these appear misleading and/or are not credible, see "Questions". The counterexample on page 4 seems to be a misunderstanding of prior work. Claims in the literature are not that diversity is *necessary* for a good ensemble. Indeed, as the submission states, an ensemble consisting of many copies of a perfect classifier are still perfect. But rather, diversity of prediction *mistakes* between models in an ensemble can provably beneficial accuracy. Without any errors, that notion is not defined. But if mistakes do occur, the variance of predictions is decreased if errors are negative correlated (e.g. (17.5) in [Alpaydin, 2014]).

* shortcomings in the experimental protocol

Several aspects about the experimental evaluation are unclear or misleading (see questions below).
- The reported accuracy value in Tables 1-3 are "highest accuracy improvements" over all epochs.
That means they are not unbiased estimates, but potentially overconfident.
- The description of ensemble elements is not clear from the text, it currently seems only provided in the Table headers.
- The specified regularization constant $\alpha=10e^{-11}$ (which should probably be $\alpha=10^{-11}$) is so small that no regularizing effect is mathematically possible even after hundreds of thousands of steps. I would expect $\alpha=0$ to work equally well.
- The exact variant of "ImageNet" dataset used is not clear. Best provide a source how the data was obtained and prepared.
- The results table lists differences in accuracy, but not absolute accuracy numbers, which would be useful to judge e.g. if the base classifiers are suitable for the task.
- It is not specified how the method's hyper-parameters were selected.
- Given the mixed positive and negative results, a test of significance should be performed if the reported results are not equally well explained by random chance (e.g. Wilcoxon signed rank).

Further comments:
- I found the analogies with human learning or research (for example the top of page 3) rather superficial, and I would recommend to remove those.
- The reference section should be corrected. Many papers are listed as arXiv or without publication venue, which actually have been published, e.g. [Arazo et al, IJCNN 2020], [Bucila etal, KDD 2006], ...

**Questions:**

* What are the key technical differences of the proposed method to simply applying gradient-based self-training to an ensemble classifier, one sample at a time?

* What are the key technical differences to prior work on ensemble self-training, such at [Ghosh, 2021]?

* Please clarify these claims: 1) page 3: "we minimize the diversity of the ensemble [...] in a way that facilitates generalization of the source training domain to the new testing domain." What does "facilitates generalization" mean here? Just higher test accuracy? In what way is lower *diversity* of the ensemble an important factor for that? 2) In the conclusion: "member models are optimized and improved like in knowledge distillation and create an upward spiral effect". What do you mean by upward spiral effect? E.g. from Figure 3 it appears that model accuracy goes up and down over epochs.

* How were the method's hyperparameters chosen, given that standard model selection is not possible in the test-time adaptation setting?

* Which variant/subset of ImageNet was used? Is the "test" data the actual "test" set or the "validation" part?

* The manuscript emphasizes a batchsize of 1, but multiple *epochs* are run on the adaptation data. Does this mean that your method must buffer all test data, or at least see it repeatedly, such that statistics can be collected? Wouldn't batch-based TTA techniques be also applicable then?

---

> ### Author Response · Authors · 2024-11-22
> **Q1-Q4**
>
> Thank you for taking the time to read our work. The issues with our experiments that you pointed out are concerning and carrying them out requires more time than the discussion period permits, so we are revising the work for a future submission. Our replies to the particular issues and questions that you gave are listed below:
>
> - You mentioned that you expected us to get the same results when we set out regularization constant to 0. We have re-ran the CIFAR-100 experiment detailed in section 4 with the regularization constant set to 0 and found that the resulting TTA ensemble learns nothing and makes the exact same predictions as a static ensemble, contrary to the expectation. We chose to use a small regularization constant simply because it was what gave the best results for us in practice. When increasing the learning rate slightly above the small one listed in the paper (say from 10^-11 to 10^10) we found that the performance gains of our TTA ensembles were significantly diminished. We are currently considering why such a small learning rate is necessary, and a more extensive parameter search is also being worked on.
>
> - Q1: Our proposed method is precisely as you describe, an application of gradient-based self-training to an ensemble classifier, one sample at a time. Although prior works on self-training typically assume that unlabeled data for adaptation and original training data are identically distributed, we extend the use-case of a rather conventional self-training technique (gradient based self-training with psuedo-labels) to scenarios where distribution shifts are present (Test Time Adaptation use case) with moderate success. No SOTA TTA records are broken with our approach, but we experimentally show that simple self-training approaches can have potential to succeed in TTA problems, which we find particularly interesting given the recently proposed issues with the Shannon Entropy as an optimization signal for TTA problems (“Entropy is not Enough for Test-Time Adaptation: From the Perspective of Disentangled Factors” Lee et al.). Although optimization of our proposed signal consequently minimizes the Shannon Entropy (Figure 1), we felt that our approach was still novel since the Shannon Entropy does not explicitly appear in our loss function (like it does in many other TTA works).
>
> - Q2: There are two key technical differences between the method proposed in “Self Training with Ensemble of Teacher Models” (Ghosh, 2021) and our work. First, the authors use one-hot encoded psuedo-labels as opposed to the soft-labels that we use, and second, the authors use an average softmax as the ensemble output, whereas we use a majority vote. Our use of soft-labels was motivated by the work “Distilling the Knowledge in a Neural Network” (Hinton, 2015), where it was found that soft-labels can provide significant information about more than a single class, unlike one-hot pseudo-labels. The use of centroid-based ensembles, like using the mean of member model outputs as the ensemble output, has been well-studied, in part because the the arithmetic mean (and other means) has nice properties (differentiable and can be easily written mathematically) for use in theorems and proofs. With the unpopularity of majority voting and our use of soft-labels, we felt that our approach was different enough from conventional self-training methods to be novel. Admittedly, more papers were read on TTA than self-training during the writing of our paper.
>
> - Q3: When we used the phrase “facilitates generalization” we meant that our method improved the performance of classifiers on data from a different distribution than the source training data. The relationship between ensemble diversity and ensemble prediction accuracy is still not well-understood, with recent work only focusing on mean-based ensembles when training and testing data are identically distributed and when models are held constant (no back propagation) - “A Unified Theory of Diversity in Ensemble Learning” (Wood, 2024). In Wood’s Bias-Variance-Diversity decomposition of the expected test error (Theorem 5, Page 10), minimizing the diversity should negatively impact the error if the bias and variance terms are held constant. However, since our TTA ensembles are neither held constant, nor use identically distributed train/test sets, nor use a mean-based combination scheme it is unclear how simple changes in the ensemble diversity should affect the remaining terms and, consequently, the expected test error. Still, we felt that our positive results were interesting compared to Wood's error decomposition.
>
> - Q4: Hyperparameters were chosen by a trial-and-error search along parameter intervals until desired test set performance was achieved. In practice, this should be impossible to achieve since we do not allow the labels needed for evaluation test set performance. This is an oversight that is currently being revised.

---

> > ### Author Response · Authors · 2024-11-22
> > **Q5-Q6**
> >
> > - Q5: For training the base models on uncorrupted data, we used the data provided in ImageNet as “train”. Corrupted data that was adapted to for IEL was taken from the “train” set provided in ImageNet-C. To evaluate the effectiveness of our adaptation to this corrupted “train” data we used data from the “test” set of ImageNet-C. This “test” set was split into two sets, one called validation_set and one called evaluation_set. Validation_set was used for hyper parameter tuning, while evaluation_set was used strictly for estimating test set errors. Each time we ran the experiment we randomly reselected the validation_set and evaluation_set samples from the larger “test” set provided in ImageNet-C.
> >
> > - Q6: You are correct, we have multiple epochs and so each unlabeled data is seen multiple times. Seeing the data points multiple times would require storage that should make batch based training possible. We plan to investigate how our approach adapts ensembles when data points are discarded after use.

---

> > > ### Comment · Reviewer_SPHA · 2024-11-26
> > >
> > > Dear authors,
> > > thank you for your clarifications. It, and the other reviews, confirm my impression that the work is currently not suitable for publication at a top tier venue such as ICLR. I maintain my recommendation of rejection.

---

### Official Review · Reviewer_ZZ56 · 2024-11-01

**Soundness:** 1
**Presentation:** 2
**Contribution:** 3
**Rating:** 3
**Confidence:** 5

**Summary:**

The authors propose a test-time adaptation (TTA) technique based on model ensembling. A set of models is simultaneously trained, and for each sample, the model with highest confidence is used as a teacher for all student models. Updates happen via standard cross-entropy. The authors show improvements over a non-adapted baseline model across CIFAR10/100-C and ImageNet-C.

**Strengths:**

The technique is simple and elegant. It generalises the typically used concept of a student/teacher setup which typically uses a single model, and is straightforward to scale in practice (by increasing the size of the ensemble).

**Weaknesses:**

**Summary**

The presentation and investigation done in the paper is well below the bar for ICLR. There are no baselines, and the results are not well presented and contextualised. The introduction to the paper is lengthy, and should be made more crisp. The contribution statement is not accurate and needs to be adapted to what is actually shown in the paper. A lot of important controls and analysis experiments are missing to back up the claims. While I think that the general idea has potential, it needs a much better investigation to be considered for acceptance. A list of actionable points is given below.

For full transparency, while I would be happy to increase the score if my points are addressed, I doubt this is doable within the rebuttal period. Depending on the other reviews, I would already suggest that the authors consider a full revision of the paper and submission of their paper to the next conference. That being said, I think this could become a very nice paper if more work is spent on the experiments, and I would be happy to iterated with the authors during the discussion period.

**Major**

**W1** There is a very lengthy introduction. The method section only starts on page 5. Before, a fair amount of related work is cited (great), but without considering any of that later as methods for comparisons.

**W2** A naiive baseline is omitted: What happens if a state-of-the-art adaptation technique like EATA, or even older/simpler techniques like TENT are used, but adapting $n$ models in parallel, and then ensembling of the outputs?

**W3** Section 3 needs a rewrite and improvements to clarity. For example, basic metrics like the cross-entropy loss are abbreviated with an extra symbol $\delta$, it would be better to clearly define the loss formulation used instead.

**W4** The contributions reference “continual learning” (l. 130, l. 134), but there is no experiment backing this up. The reference should be removed.

**W5** Claim 3 in the contributions (ll. 135-136) states that TTA requires a full batch. This is misleading or even wrong. There are certainly techniques that measure improvements when only a single sample is available (= model is adapted on that sample, and performance is measured) before the domain changes (e.g. Batch norm adaptation, or MEMO). However, in the setting considered here, model updates still accumulate on the same domain. In that setting, any TTA technique, like TENT, can be made suitable for the discussed setting by only updating every N steps (where N is the batch size), collecting the samples in the meantime.

**W6** Claim 4 in the contributions is not at all corroborated by results in the paper and should be dropped.

**W7** The paper needs to add recent methods to compare to. The current tables provide a lot of irrelevant details, and should be compressed. Instead of listing all different domains, it would be better to run a sufficient number of baseline models on the considered datasets to contrast the results. When doing so, it could be interesting to investigate whether the ensembling mechanism proposed is *orthogonal* to other adaptation methods: E.g., if we consider the EATA loss function for ensembling, does this improve over EATA?

**W8** Analysis should be added: What happens if the number of models in the ensemble varies? How robust is this technique to common problem in cross-entropy based adaptation (model collapse) when training over long time intervals?

**W9** In case the authors decide to keep the continual learning claim: How does the model perform in the continual adaptation settings used in CoTTA or EATA, and how does the model perform on long-term adaptation on CCC (Press et al., 2023)?

**Minor**

- Related work in l. 178: Batch norm adaptation (Schneider et al., 2020) re-estimates batch norm statistics, and this was shown to also work on single samples. Test time training (Sun et al., 2019) also considers single samples. TENT, in contrast, *additionally* performs entropy minimisation akin to the cross-entropy loss discussed in the study.
- Figure 1 misses a legend. The color code could be adapted, and e.g. corruptions of the same type could get the same color with different marker colours. That would make the plot more interpretable than the current 15 random colours.

**Questions:**

Please see the weaknesses above that are directly actionable. No further Qs.

---

> ### Author Response · Authors · 2024-11-22
>
> Thank you for taking the time to read our work and give your review, we are grateful that you think our work has potential. In light of the glaringly absent experiments that you pointed out, we are deciding to revise the paper for a future submission. All of the clarity and technical issues with that writing that you mentioned are being worked on now (W1, W3, W4, W6). Our replies the particular issues you raised are given below:
>
> - W2: Before we got to the bulk of the experiments detailed in the paper we tried using a naive baseline as you described: an ensemble where each individual model has TENT applied to it in parallel. In these experiments we found that we consistently underperformed compared to the naive baseline, but felt that the core of our novelty was in that 1) we make progress on TTA problems using non-standard methods (ensemble pseudo-labeling is popular in self-training, but less so in TTA, and the Shannon entropy signal makes no appearance in our loss function)  and 2) we enable models to learn from each other when incoming data is unlabeled and has its distribution shifted from that of the original training data.
>
> - W5: This is a good observation that we overlooked. Going forward, we will emphasize that by using multiple epochs we inherently allow storage of incoming data, and so any methods that utilize batch calculations should also be usable given our assumptions.
>
> - W7: Going forward we plan to carry out more comprehensive experiments that relate IEL to other popular TTA methods. In particular, it seems that IEL can be used in tandem with many TTA methods that depend on optimization of only batch normalization layer parameters, like TENT and EATA. Could you say more about what you mean by “if we consider the EATA loss function for ensembling”?
>
> - W8: Due to computational limitations at our public university, we were not able to re-run our experiments while varying the number of models in the ensemble. For similar reasons, we were also not able to investigate the long run behavior of our approach. We are currently working on running experiments for both of these issues.

---

> > ### Comment · Reviewer_ZZ56 · 2024-11-26
> >
> > Dear authors, please let me know if you are still revising the paper before the revision deadline -- I will stay put an remain happy to re-evaluate.
> >
> > W2: It sounds like this should still be reported and analysed then.
> >
> > W5: Thanks for acknowledging. It would be good to keep this in mind when deciding on a good set of baselines.
> >
> > W7: With my comment, I meant that it might be possible to replace your particular formation of the loss with any self-learning loss, as the ensembling method is orthogonal to the self-training loss as far as I can judge. For ETA/EATA in particular, it might be interesting to convert the loss function and make it suitable for application in your framework. I see this more as an optional step though, which might get necessary if your formulation (without the bells and whistles in the EATA framework) does not outperform EATA -- so this point is optional.
> >
> > W8: I think this would add a lot to the paper, thanks for considering. Note that performance on ImageNet-C can be deceptive, and some models that appear to work on ImageNet-C then collapse when running on e.g. CCC (Press et al., [Neurips 2023](https://proceedings.neurips.cc/paper_files/paper/2023/hash/7d640f377893fc5f22b5610e175ef7c3-Abstract-Conference.html)).

---

### Official Review · Reviewer_LBnv · 2024-11-04

**Soundness:** 1
**Presentation:** 2
**Contribution:** 2
**Rating:** 1
**Confidence:** 4

**Summary:**

The paper introduces an intra-model ensemble learning method for single sample test-time adaption. It minimizes the cross-entropy losses between the output with the highest confidence score and all other classifier's outputs. It optimizes all trainable parameters (except for BN layers) and requires only a single sample for TTA. It achieves improved performance on corrupted datasets including CIFAR10-C, CIFAR100-C and ImageNet-C.

**Strengths:**

- The paper addresses a challenging TTA settings where only a single, unlabeled sample is given for adaptation during test time.
- The paper adopts an interesting approach of ensemble learning to dynamically optimize a group of learners (pre-trained models), showing improved TTA performance.

**Weaknesses:**

- The proposed algorithm offers no substantial improvement over existing ensemble learning methods. It simply combines 1) selecting the most confident prediction and 2) cross-entropy minimization of ensemble models. Technical contributions to both ensemble learning and single-sample TTA remain limited.
- The paper lacks sufficient experimentation to demonstrate the proposed method’s effectiveness. It only compares results across different backbone architectures, without considering other baseline methods suitable for single-sample TTA, such as NOTE [1] and REALM [2]. Additionally, it does not explore alternative TTA settings, such as continual TTA, where incoming domains continuously change.
- The experiment section (Sec. 4) requires more careful writing and clarification. For instance, it should include a clear definition and detailed description of the tuning set samples, as well as more comprehensive experiments, including ablation studies, to examine the correlation between the number of ensemble models and TTA performance.
- In Section 4.1, the authors state that no catastrophic forgetting was observed on the ImageNet-C dataset. However, this is unlikely to be accurate since only 7,000 samples per corruption type from ImageNet-C were used for evaluation. More rigorous experiments and substantiated claims are needed.
- As noted in the limitations, the proposed method requires significant computational resources to utilize multiple pre-trained networks. However, the paper does not provide any empirical analysis or comparison of computational cost or adaptation latency.

[1] Gong et al., Note: Robust continual test-time adaptation against temporal correlation. NeurIPS'22. \
[2] Seto et al., REALM: Robust Entropy Adaptive Loss Minimization for Improved Single-Sample Test-Time Adaptation. WACV'24.

**Questions:**

Please refer to the weaknesses part.

**Details Of Ethics Concerns:**

There is no potential violation of the CoE.

---

> ### Author Response · Authors · 2024-11-22
>
> Thank you for taking the time to read and review our work.
> Our responses to your review are listed below:
>
> 1. Although IEL is rather simple, we believe its simplicity is a strong feature. Our method does not beat existing ensemble or TTA algorithms, but we feel our novelty is in that we uniquely train a set of models, unlike in many standard TTA methods, by having the models learn from each other in a closed system. We find that IEL resultingly minimizes the Shannon Entropy of member model outputs (Figure 1), which is a commonplace optimization signal in many TTA methods, while relying explicitly on an Ensemble-based optimization signal.
>
> 2. Due to computational restrictions at our public university we were unable to carry out certain experiments. We will attempt to compare the NOTE and REALM methods to ours. In the single-sample setting, both NOTE and REALM inherently depend on batch statistics, whereas we do not. Going forward, we will leave room for comparisons to SOTA TTA methods like NOTE and REALM. We will update this comment as we get results.
>
> 3. We will add a clear and detailed description of the tuning set samples used in our experiments. The correlation between the number of ensemble models and TTA performance was not more rigourously tested due to computational restraints.
>
> 4. We agree that a larger subset of ImageNet-C is required to accurately assess the performance gains given by IEL. Our computational resources are limited in our academic university lab, unlike at larger companies, but this is a good suggestion that we will proceed with over time.

---

> > ### Comment · Reviewer_LBnv · 2024-11-27
> >
> > Thank you for your response.
> >
> > The reviewer acknowledges that the proposed method has its unique feature of intra-model ensemble learning, which is simple yet straightforward. It would be a great approach to tackle TTA task if its algorithmic details and empirical studies are more thoroughly elaborated.
> >
> > Additionally, the reviewer suggests that the authors consider using a small-sized pre-trained model to conduct experiments, especially if computing resources are matter. Since the paper has mainly focused on the single sample adaptation setup (batch size of 1), it would be possible to experiment on the large-scale dataset (e.g., ImageNet) with pre-trained model of moderate sizes.
> >
> > Once again, the reviewer appreciates the authors' responses.

---

### Meta-Review · Area_Chair_aMLN · 2024-12-21

**Metareview:**

Test-time adaptation aims to update a model online given test data to maintain or improve accuracy and other task metrics. This work adapts not a single model, as is usually the case, but a set of models that is applied as an ensemble. For prediction the majority vote is taken, then for adaptation all the models are updated by cross-entropy with the highest probability output for the majority class as the target. While it is common for test-time adaptation methods to require batches or at least sequences of test inputs, this method can reduce generalization error given a single point at a time (which has been addressed by prior work, such as MEMO and SAR, but is still distinctive).  The experiments evaluate on the standard benchmarks for this topic of ImageNet and CIFAR-10/100 with corruptions (ImageNet-C, CIFAR-100-C, CIFAR-10-C). The results show consistent improvement over static ensembles without test-time updates.

Strengths:

- Test-time adaptation is a popular topic where progress is being made, and this work focuses on 1. a particularly challenging and relevant setting of single input adaptation (LBnv, SPHA, 4Chx) and 2. ensembling which is understudied for this purpose (LBnv, 4Chx)
- The method is simple and scalable since the size of the ensemble can be varied (ZZ56), as can the size of models in the ensemble (LBnv).

Weaknesses:

- The experiments compare to ensembling, but lack baselines for test-time adaptation, which are the natural comparisons (LBnv, ZZ56, SPHA, 4Chx). Without evaluating existing adaptation methods, and ideally measuring multiple settings like episodic and continual test-time adaptation (LBnv, 4Chx), it is difficult or impossible to gauge the improvement due to the proposed method.
- Test-time adaptation of an ensemble requires the computational overhead of inferring and updating the set of models in the ensemble. However, this computational overhead is neither measured, discussed, or mitigated (LBnv).
- The claims and the evidence are disconnected in the results of this work or in its exclusion of prior works (ZZ56, SPHA). This includes the existence of single input adaptation methods (like BN, MEMO), the study of continual settings by CoTTA/EATA/CCC, and the off-topic and unsubstantiated claim of relevance to human collaboration (stated by the 4th bullet point of the introduction).

Rationale for decision:

Four expert reviewers agree on rejection (LBnv: 1, ZZ56: 3, SPHA: 3, 4Chx: 3). The meta-reviewer agrees with the raised weaknesses, and would like to highlight the missing baselines such as parallel adaptation of multiple models and the missing comparisons to existing work. Given the important omissions in the experiments and related work, the weaknesses are more important than the strengths, and the meta-reviewer sides with rejection.

**Additional Comments On Reviewer Discussion:**

The authors respond to each review, and the reviewers reply in turn to engage in discussion and confirm their evaluations. The authors do not further reply or revise the submission, but they do comment "we are deciding to revise the paper for a future submission" although the submission is not withdrawn.

---

### Decision · Program_Chairs · 2025-01-22

Reject